Open access | Protocol

# Suicide and self-harm by burns in Pakistan: a scoping review protocol

Aisha Noorullah ,[1] Shahina Pirani ,[1] Emily Bebbington ,[2] Murad Khan [1]

¹Department of Psychiatry, Aga Khan University, Karachi, Pakistan
²Centre for Mental Health and Society, Bangor University, Wrexham, UK

**Correspondence to**
Dr Emily Bebbington;
e.bebbington@bangor.ac.uk

## ABSTRACT

**Introduction** Suicide is a global public health problem. Self-inflicted burns are one of the most severe methods of suicide, with high morbidity and mortality. Low-income and middle-income countries contribute 40% of all suicidal burns. Pakistan lacks comprehensive burns surveillance data, which prevents an understanding of the magnitude of the problem. This scoping review aims to understand the scope of the problem of suicide and self-harm burns in Pakistan and to identify knowledge gaps within the existing literature related to this specific phenomenon.
**Methods and analysis** This scoping review will follow the methodological framework proposed by Arksey and O'Malley. We will search electronic databases (PubMed, Cochrane, Google Scholar and Pakmedinet), grey literature and a reference list of relevant articles to identify studies for inclusion. We will look for studies on self-inflicted burns as a method of suicide and self-harm in Pakistan, published from the beginning until December 2023, in the English language. Two independent reviewers will screen all abstracts and full-text studies for inclusion. The data will be collected on a data extraction form developed through an iterative process by the research team and it will be analysed using descriptive statistics.
**Ethics and dissemination** Ethical exemption for this study has been obtained from the Institutional Review Board Committee of Aga Khan University Karachi, Pakistan. The findings of the study will be disseminated by conducting workshops for stakeholders, including psychiatrists, psychologists, counsellors, general and public health physicians and policymakers. The findings will be published in national and international peer-reviewed journals.

## BACKGROUND

Burns are a global public health problem, causing an estimated 180 000 deaths per year.[1] Mortality from burn injuries is higher in low-income and middle-income countries (LMICs). Almost two-thirds of deaths occur in the WHO African and Southeast Asia regions.[1]

Self-inflicted burns are one of the most painful and violent methods of suicide, with high morbidity and mortality.[2] Suicide by burning is a rare condition in high-income countries (0.06%–1% of all suicides) but is more frequent in LMICs (accounting for as many as 40.3% of all suicides).[3] 77% of all suicides occurred in LMICs in 2019.[4]

## STRENGTHS AND LIMITATIONS OF THIS STUDY

⇒ The Preferred Reporting Items for Systematic Reviews and Meta-Analyses protocol and scoping review reporting guidelines have been followed in the preparation of this manuscript and will be followed throughout the scoping review process.
⇒ The review will be conducted in close collaboration between all authors. Any disagreement about the inclusion of studies will be discussed among the team until a consensus is reached. This process increases the validity and reliability of the review.
⇒ Academic librarians will be consulted to develop a search strategy and identify relevant papers in different databases.
⇒ Only English language studies will be included. This is the dominant language used for the practice of medicine in Pakistan, but it may restrict the breadth of our results.

Burn injuries can result in prolonged hospitalisation and long-term physical and psychological sequelae.[5] Non-fatal burns contribute to morbidity, such as disfigurement and disability, leading to stigma and abandonment for the survivors.[1] Burns are among the leading causes of disability-adjusted life years lost in Asia.[6]

In Pakistan, a South Asian Islamic country, there are no official statistics on suicides at the national level.[7] The WHO estimated there were 19 331 suicidal deaths and a rate of 8.9/100 000 people in 2019.[4] Until recently, suicides were treated as criminal acts in Pakistan, a religiously condemned and highly stigmatised. In December 2022, suicide and self-harm acts were decriminalised.[8] However, it will take time for the abolition of the legal processes associated with them. Suicide and self-inflicted injuries are under-reported due to religious, sociocultural and medico-legal processes. Suicidal burn injury is an important but under-researched area in Pakistan. Due to fear of societal stigma and legal issues, families withhold information about assault and self-harm and label it as accidental. This cany lead to misclassification and undercounting of cases.[9 10]

There are few systematic reviews and meta-analyses on suicidal burns from various parts of the world.[11–14] These reviews revealed that the highest prevalence of suicidal burns is found in South Asian and Eastern Mediterranean countries, including India, Iran, Iraq, Afghanistan and Sri Lanka. In these studies, women represent many cases of suicidal burning with high mortality. Women who engage in suicidal self-burning are typically young, married and mothers.[2 11 12 14]

A sysematic scoping review has been conducted on the terminology and methods used to differentiate the intent of hospitals presenting burn injuries in South Asia.[15] However, there is no study that we are aware of that systematically investigates the demographics and outcomes of all suicidal burns in Pakistan.

The importance of reviewing the existing literature on burn studies in Pakistan lies in investigating the true extent of the burden imposed by suicidal burn injuries. It is crucial to explore the frequency, demographics and outcomes reported to gain insights into this issue. Furthermore, the comparability of the findings in the literature remains undetermined, emphasising the need for a scoping review to assess the availability and comparability of the existing research. Identifying specific groups that are prominently affected by these injuries may also highlight the necessity for further focused studies to address their special needs. The gravity of burn injuries in terms of fatality[16] and the availability of the method, with their profound impact on physical and mental health, differentiate these cases from other forms of self-harm. This necessitates a comprehensive understanding of the under-researched subject matter in Pakistan. This scoping review will also serve as an initial step as part of the planning to commence a surveillance system in Pakistan by gauging the magnitude of the problem and its implications.

A search was conducted at the Open Science Framework (OSF), Figshare, PubMed and Prospero for systematic and scoping reviews on self-inflicted burn injuries in Pakistan, but no prior research specific to Pakistan was identified.

## Aims and objectives

This scoping review aims to understand the scope of the problem of suicide and self-harm using burns as a method of attempt across various age groups in Pakistan. It will also provide evidence to identify the knowledge gap in the existing literature on the phenomenon of interest. The specific objectives of the review include:

1. To assess the availability of data on the frequency of self-harm and suicide cases involving burns across various demographic factors in Pakistan.
2. To examine the methods commonly used for self-inflicted burns, including self-harm and suicide in Pakistan.
3. To evaluate the outcomes of self-inflicted burns in Pakistan.

## METHODS

We will conduct a scoping review to synthesise evidence on self-inflicted burns as a method of self-harm or suicide in Pakistan. This protocol has been written using the Preferred Reporting Items for Systematic Reviews and Meta-Analyses extension for Scoping Reviews (PRISMA-ScR) (refer online supplemental file 1). This scoping review will follow the methodological framework proposed by Arksey and O'Malley[17] and amended by Levac *et al*.[18] This framework consists of six stages: (1) identifying the research question, (2) identifying relevant studies, (3) study selection, (4) data charting, (5) data analysis and reporting the results and (6) consultation.

The scoping review method is described, considering the above six stages. This review has been registered with the Open Science Framework: https://doi.org/10.17605/OSF.IO/XA68S.

### Stage 1: identifying the research question

Based on the literature review, we formulated the following research questions.

#### Research question 1A

What is the frequency of self-harm and suicide cases involving burns in Pakistan?

#### Research question 1B

How is the frequency of self-harm and suicide cases involving burns distributed across different demographic factors such as age, sex and marital status in Pakistan?

#### Research question 2A

What are the common methods used for self-inflicted burns, including self-harm and suicide, in Pakistan?

#### Research question 2B

Are there any variations in the methods of self-inflicted burns based on demographic factors such as age, sex and marital status in Pakistan?

#### Research question 3A

What are the short-term and long-term outcomes of self-inflicted burns in Pakistan?

#### Research question 3B

Are there any variations in the outcomes of self-inflicted burns based on demographic factors such as age, sex, and marital status in Pakistan?

### Stage 2: identifying relevant studies

To identify relevant studies, an academic librarian will assist in developing a robust search strategy. We will identify MESH keywords for each concept (burns, suicide, self-harm and intentional) included in the review question and use synonyms of key terms. Additionally, we will use Boolean operators with a combination of terms including (self-burning) OR (self-immolation) OR (burns) OR (self-mutilation) AND (self-harm) OR (intentional) OR (deliberate self-harm) AND (suicide) OR (attempted

**Table 1** Inclusion and exclusion criteria for the scoping review

| Themes | Inclusion criteria | Exclusion criteria |
|---|---|---|
| Population | Studies included populations with suicidal burns, in all genders and across all ages. | Population with unintentional burn injuries. |
| Concept | ► Studies focused on burns in general, where the intent was self-harm or suicide.<br>► Studies focused on self-harm or suicide, where burns were used as one of the methods.<br>► Articles using different terminologies for suicidal burns (eg, self-mutilation, self-immolation and self-inflicted burns) | Any article on burns that has no connection with suicide or self-harm. |
| Context | ► Studies conducted within the geographical boundaries of Pakistan.<br>► Studies conducted at hospital or community setting.<br>► Cases may be based on autopsy reports, forensic reports or medicolegal reports. | Studies on Pakistanis residing outside the country. |
| **Additional filters** | | |
| Article type | ► Original research article with any study design, quantitative or qualitative study.<br>► Case studies, case series or case report<br>► Editorial and short communication<br>► Systematic, scoping or rapid review.<br>► 'Grey literature' that includes unpublished theses and dissertations. | Media news |
| Reporting/language | ► Articles published in the English language. | Other languages |
| Date of publication | Studies published from the beginning until December 2023 will be considered for this review. | Studies published after December 2023 |

suicide) OR (parasuicide) AND (Pakistan) to look for relevant literature (refer online supplemental file 2).

We will search four databases, including the National Library of Medicine's MEDLINE (PubMed), Google Scholar, Cochrane Library and Pakmedinet. We will also search 'grey literature' that includes unpublished theses, conference papers and other reports. In addition, we will employ an ancestry approach, checking the reference lists of retrieved articles for relevant references. We will search databases from the beginning until December 2023, published in the English language. The search strategies for each database will be similar in structure, with similar search terms and synonyms. Additional keywords and sources identified during the process will be noted and reported. We will keep a separate record of the search results in a spreadsheet file, including the date of search, search terms and the number of retrieved articles. The search results will be exported into the reference management software Endnote, and duplicate articles will be removed.

### Eligibility criteria

The selection of studies in the review will be based on the thematic framework 'Population–Concept–Context (PCC)' recommended by the Joanna Briggs Institute for scoping reviews,[19] outlined under broader eligibility criteria. Consideration would be given to additional filters for type of publication and language preference (refer table 1)

### Stage 3: study selection

Two researchers will conduct an exercise before initiating the search to ensure reliability in selecting articles for inclusion. This exercise will involve the independent screening of a random sample of 5% of the citations by the two independent reviewers. If the two researchers show low agreement, they will revisit the selection criteria.

The study selection process involves two steps. In the first step, two reviewers (SP and AN) will screen the study titles and abstracts to determine eligibility for full-text screening based on predefined inclusion criteria. A spreadsheet file will be used to record the search results, including the search date, terms used and the number of retrieved articles. After exporting the search results into the Endnote reference management software, duplicate articles will be removed. In the second stage of the selection process, the full text of the articles will be retrieved and uploaded as an endnote. These will undergo independent screening by the same two reviewers (SP and AN), and any disagreements in eligibility between the reviewers will be resolved through discussion and involving a third researcher (MK).

We will use an adapted version of the PRISMA flow diagram to report the final numbers of the included studies. At the full-text screening stage, we will record the reasons for excluding the studies.

### Stage 4: charting the data

The data charting form has been developed in a Microsoft Excel sheet after consultation with the subject expert

and by reviewing the existing literature (refer online supplemental file 3). This form included information on the source of the study, the place where the study was conducted, study design, time of the study, number of burn cases, number of suicidal burns, mechanism of burns, age, gender, marital status, reasons for burns and outcome. We will pretest the form and modify it accordingly.

### Stage 5: data analysis and reporting the results

We will analyse the data descriptively using a data charting form. The study question included the frequency of suicidal burns across various demographics, methods commonly used for self-inflicted burns and the outcome of injury, which will be the basis for analysis. We will present the characteristics of the study related to suicidal burns using tabulations or figures/flow charts, where appropriate, to present a synthesis of key findings according to the scoping review objectives.

The researchers will report the scoping review according to the PRISMA-ScR checklist.

### Stage 6: consultations

In this final stage, we will consult with the subject experts (MK and EB) to validate the findings. The subject experts will help to situate the findings in the context of broader research, policy, and practice.

Our scoping review work is in the study selection phase, which began in November 2023. The authors anticipate that this review will be completed and submitted for publication in the latter half of 2024.

### Patient and public involvement

There has been no patient or public involvement in the development of this protocol, nor will there be in the scoping review itself.

### Coordination, monitoring and quality control

The study will ensure quality control by adhering to the standard protocols for conducting scoping reviews. This includes:

▶ Use of the five-stage framework by Arksey and O'Malley for conducting a literature review.
▶ Development of a comprehensive search strategy.
▶ Training and debriefing session for the two staff on conducting the scoping review.
▶ Independent review by two experienced team members.
▶ Frequent meetings with the research team to review the findings.

### ETHICAL CONSIDERATIONS AND DISSEMINATION

Ethical exemption for this study has been obtained from the Institutional Review Board Committee of Aga Khan University Karachi, Pakistan. We will disseminate the findings of the study by conducting workshops for stakeholders, including psychiatrists, psychologists, counsellors, general and public health physicians and policymakers. Additionally, we will publish the findings in national and international peer-reviewed journals.

**Contributors** AN and MK conceived the study and were involved in concept mapping and the planning of the study. SP looked for a literature review and developed a study tool. SP and AN contributed to writing the study protocol. EB critically reviewed the protocol. MK intellectually contributed to the study. All authors have read and approved the final manuscript.

**Funding** This work was supported by the South Asia Self-Harm Initiative. The South Asia Self-Harm initiative is funded by the Global Challenges Research Fund, UK Research and Innovation. Project number: MR/P028144/2. Principal investigator: Professor Catherine Robinson. The funder has not been involved in the development of this protocol.

**Competing interests** None declared.

**Patient and public involvement** Patients and/or the public were not involved in the design, or conduct, or reporting, or dissemination plans of this research.

**Patient consent for publication** Not applicable.

**Provenance and peer review** Not commissioned; externally peer reviewed.

**ORCID iDs**
Aisha Noorullah http://orcid.org/0000-0001-5755-7609
Shahina Pirani http://orcid.org/0000-0001-6908-9435
Emily Bebbington http://orcid.org/0000-0003-1332-7558
Murad Khan http://orcid.org/0000-0002-5034-6865

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
