## [Reviewer comments · BMJ Open]

This paper was submitted to a another journal from BMJ but declined for publication following peer review. The authors addressed the reviewers' comments and submitted the revised paper to BMJ Open. The paper was subsequently accepted for publication at BMJ Open.

ARTICLE DETAILS

TITLE (PROVISIONAL)	Suicide and Self-Harm by Burns in Pakistan: A Scoping Review Protocol
AUTHORS	Noorullah, Aisha; Pirani, Shahina; Bebbington, Emily; Khan, Murad

VERSION 1 – REVIEW

REVIEWER	Abrams, Thereasa Sam Houston State University
REVIEW RETURNED	23-Oct-2023

GENERAL COMMENTS	This paper is more of an outline than a manuscript, with no results presented. Having written an article on suicide by burning in the U.S., it seems important to review all literature regarding suicide by burning globally, particularly since there are very few and there appears to be no references for articles relative to Pakistan. This paper presents a well planned design for a systematic review however, the authors should report their findings from the proposed systematic review.
--

REVIEWER	Ghossoub, Elias American University of Beirut Medical Center
REVIEW RETURNED	29-Oct-2023

GENERAL COMMENTS	This is an well-written manuscript that describes in detail the protocol for a scoping review on suicide and self-harm by burning in Pakistan. I have several comments: _ One of the main issues with this project is the presence of another scoping review (Ref. 17) that has been published and includes Pakistan among the countries within its scope. The authors should clarify how their project is different from Ref. 17._ The authors do a good job in outlining the eligibility criteria for studies. Will they include studies that focus on burn injuries in general as well as studies that focus on suicide/self-harm? Both types of studies may include valuable data about self-harm/suicide by burning._ The authors exclude studies published in languages other than English. Is it possible that they may miss out on local studies published in Urdu?_ The authors should use the active voice instead of the passive voice in the methodology section._ The authors' reference (7) is outdated.
---

	_ The introduction section needs to be better organized and focused. The section on Pakistan's profile is comprehensive but is only marginally connected to the objectives of the study.
--	--

VERSION 1 – AUTHOR RESPONSE

Reviewer: 1

Dr. Thereasa Abrams, Sam Houston State University

Comments to the Author:

This paper is more of an outline than a manuscript, with no results presented. Having written an article on suicide by burning in the U.S., it seems important to review all literature regarding suicide by burning globally, particularly since there are very few and there appears to be no references for articles relative to Pakistan. This paper presents a well planned design for a systematic review however, the authors should report their findings from the proposed systematic review.

Editor's note: Please note that some of reviewer 1's comments are not relevant to your manuscript as it is a protocol.

Thank you for taking the time to review this manuscript and for your helpful comments. We have addressed them where possible, but please note this is a protocol paper so we do not yet have any results to share. We hope you feel the manuscript merits publication. We have added additional detail to our introduction about the global studies on suicidal burns.

Reviewer: 2

Dr. Elias Ghossoub, American University of Beirut Medical Center

Comments to the Author:

This is a well-written manuscript that describes in detail the protocol for a scoping review on suicide and self-harm by burning in Pakistan. I have several comments:

Thank you for taking the time to review this manuscript and for your helpful comments. We have addressed each in detail, which we believe has improved the manuscript. We hope you can now support its publication.

One of the main issues with this project is the presence of another scoping review (Ref. 17) that has been published and includes Pakistan among the countries within its scope. The authors should clarify how their project is different from Ref. 17.

We have revised the background section, so reference number 17 is now reference number 15. There are a number of differences between reference 15 and this review. We felt that these were best illustrated in the following table:

Reference number 17 study (now its ref. 15) (Bebbington et al., 2023)	Present study
- The objective of the study is to understand the terminology and methods used to differentiate the injury intent of hospital burn patients in South Asia.	- The objective of the study is to understand the scope of the problem of suicide and self-harm using burns as a method of attempt across various demographics (age, gender, education,

	marital status, socio-economic status, and others) in Pakistan.  - Methods commonly used for suicidal burns in Pakistan. - Evaluate the outcomes (mortality) of suicidal burns in Pakistan.
 - Those articles were included where data has been collected only from hospital burns patients. - Studies conducted in South Asia i.e. Bangladesh, Bhutan, India, Sri Lanka, Maldives, Nepal, and Pakistan. - 	 - All those suicidal burns studies will be included where data has been collected from hospital or community settings, Police reports, Autopsy/Forensic studies, and Medicolegal reports. - Studies conducted within Pakistan.
 - Included studies with burns injury intent (e.g. unintentional, intentional, accidental, homicidal, suicidal, undetermined). 	 - Include burns studies with the intent of suicide only.

We have added differences in the manuscript.

The authors do a good job in outlining the eligibility criteria for studies. Will they include studies that focus on burn injuries in general as well as studies that focus on suicide/self-harm? Both types of studies may include valuable data about self-harm/suicide by burning.

Yes, we will include both types of studies: those exclusively focused on burns as methods of suicide or self-harm, and those studies where burns are used as one of the methods of suicide and self-harm.

We have added following in the Inclusion criteria:

- *“Studies that explicitly focused on burns as a method of self-harm or suicide.*

-*Studies focused on self-harm and suicide where burns are used as one of the methods.”*

The authors exclude studies published in languages other than English. Is it possible that they may miss out on local studies published in Urdu?

To the best of the researchers' knowledge, almost all studies conducted in Pakistan on suicidal behaviours are published in English. We have acknowledged that this may be a limitation in the 'Strengths and limitations' section:

Strengths and limitations: *“Only English language studies will be included. This is the dominant language used for the practice of medicine in Pakistan, but it may restrict the breadth of our results.”*

he authors should use the active voice instead of the passive voice in the methodology section.

We have changed the methodology section to the active voice. Please refer to the revised manuscript if necessary.

_ The authors' reference (7) is outdated.

Reference number 7 deleted and new references have been added.

The introduction section needs to be better organized and focused. The section on Pakistan's profile is comprehensive but is only marginally connected to the objectives of the study.

Thank you for the valuable comment. We have revised the Introduction section.

Reviewer: 1

Competing interests of Reviewer: I have published an article focused on suicide by burning in the U.S. using CDC National Violent Death Reporting System data.

Reviewer: 2

Competing interests of Reviewer: None

VERSION 2 – REVIEW

REVIEWER	Ghossoub, Elias American University of Beirut Medical Center
REVIEW RETURNED	26-Dec-2023
GENERAL COMMENTS	I thank the authors for adequately addressing my comments. I would respectfully suggest that they reconsider outright excluding articles that are focused on burns in general. These articles may assess the context of burn injuries and include suicide/self-harm as a context/variable. Excluding them without doing a full-text screening may lead to the authors missing out on important information.

VERSION 2 – AUTHOR RESPONSE

Reviewer: 2

Dr. Elias Ghossoub, American University of Beirut Medical Center

Comments to the Author:

I thank the authors for adequately addressing my comments. I would respectfully suggest that they reconsider outright excluding articles that are focused on burns in general. These articles may assess the context of burn injuries and include suicide/self-harm as a context/variable. Excluding them without doing a full-text screening may lead to the authors missing out on important information.

We apologies for any confusion. For more clarity, we have re-worded as:

Studies focused on burns in general, where the intent was self-harm or suicide.

(added in the inclusion criteria of the manuscript)

It means that we will include articles that focus on burns in general, and we will conduct full-text screening to determine the intention of burns as suicide/self-harm.

Reviewer: 2

Competing interests of Reviewer: None